# Impact of COVID-19 Pandemic on Frontline Pembrolizumab-Based Treatment for Advanced Lung Cancer

**DOI:** 10.3390/jcm12041611

**Published:** 2023-02-17

**Authors:** Tawee Tanvetyanon, Dung-Tsa Chen, Jhanelle E. Gray

**Affiliations:** Department of Thoracic Oncology, H. Lee Moffitt Cancer Center and Research Institute, Tampa, FL 33612, USA

**Keywords:** lung cancer, pembrolizumab, immunotherapy, COVID-19

## Abstract

Background: Pembrolizumab monotherapy or pembrolizumab plus chemotherapy has become an important frontline treatment for advanced non-small cell lung cancer (NSCLC). To date, it remains unclear how the coronavirus disease 2019 (COVID-19) pandemic impacted the treatment outcome. Methods: A quasi-experimental study was conducted based on a real-world database, comparing pandemic with pre-pandemic patient cohorts. The pandemic cohort consisted of patients who initiated treatment from March to July 2020, with follow-up through March 2021. The pre-pandemic cohort consisted of those initiating treatment between March and July 2019.The outcome was overall real-world survival. Multivariable Cox-proportional hazard models were constructed. Results: Analyses included data from 2090 patients: 998 in the pandemic cohort and 1092 in the pre-pandemic cohort. Baseline characteristics were comparable, with 33% of patients having PD-L1 expression level ≥50% and 29% of patients receiving pembrolizumab monotherapy. Among those treated with pembrolizumab monotherapy (N = 613), there was a differential impact of the pandemic on survival by PD-L1 expression levels (*p*-interaction = 0.02). For those with PD-L1 level < 50%, survival was better in the pandemic cohort than the pre-pandemic cohort: hazard ratio (HR) 0.64 (95% CI: 0.43–0.97, *p* = 0.03). However, for those with PD-L1 level ≥ 50%, survival was not better in the pandemic cohort: HR 1.17 (95% CI: 0.85–1.61, *p* = 0.34). We found no statistically significant impact of the pandemic on survival among patients treated with pembrolizumab plus chemotherapy. Conclusions: The COVID-19 pandemic was associated with an increase in survival among patients with lower PD-L1 expression who were treated with pembrolizumab monotherapy. This finding suggests an increased efficacy of immunotherapy due to viral exposure in this population.

## 1. Introduction

Pembrolizumab, a PD-1 checkpoint inhibitor, has become a common frontline treatment for patients with advanced non-small cell lung cancer (NSCLC). Pembrolizumab monotherapy produces a survival outcome that is better than chemotherapy for patients with tumors expressing PD-L1 level ≥ 50% [1,2]. In addition, pembrolizumab plus platinum-based chemotherapy has been shown to be superior to chemotherapy alone, regardless of PD-L1 expression level [3]. Pembrolizumab monotherapy has now received a regulatory approval from the U.S. Food and Drug Administration for treatment of tumors expressing PD-L1 ≥1%. Meanwhile, in 2020, the COVID-19 pandemic, caused by the severe acute respiratory syndrome coronavirus-2 (SARS-CoV-2), spread rapidly into the United States. Data from the Centers for Disease Control and Prevention (CDC) indicated that, while in the first week of March 2020, fewer than 100 infected cases per day were recorded, by the end of March 2021, over 30 million total cases were recorded [4]. It has been speculated that the actual number of infected cases may be substantially higher than the CDC data due to under-reporting of cases and shortage of testing kits. A seroprevalence study utilizing SARS-CoV-2 antibody conducted in February 2022 estimated that nearly 60% of the US population had been infected by the virus [5].

The COVID-19 pandemic may have affected cancer patients in several ways. First, during the pandemic, medical resources may have been diverted to COVID-19-related needs, resulting in a difficult access to cancer care and delayed treatment [6]. Second, cancer patients are more susceptible to SARS-CoV-2 infection and may develop more severe symptoms, thus increasing the risk of morbidity and mortality during cancer treatment [7,8]. Third, SARS-CoV-2 infection can modulate immune response, and this may alter the safety or the efficacy of pembrolizumab-based treatment [9,10]. Available studies have shown that, among cancer patients, factors such as having multiple comorbid diseases or cytotoxic chemotherapy, but not immunotherapy, can increase the risk of death following COVID-19 infection [11,12]. To date, toxicities of checkpoint inhibitors do not appear to have increased during SARS-CoV-2 infection or vaccination [13,14].

Understanding the differential effect of SARS-CoV-2 infection on cancer treatment may help the oncology community better prepare for future pandemics. Since advanced lung cancer is a prevalent disease and pembrolizumab is a commonly used treatment for this indication, the information may have a significant public health implication. Because a randomized study will not be feasible for this purpose, we utilized a large, population-based database to conduct a quasi-experimental study, with the pandemic serving as a natural experiment. We characterized the impact of the COVID-19 pandemic on the survival of advanced NSCLC patients during pembrolizumab-based regimens by comparing patient cohorts in the year before and the year of the pandemic. We also investigated diverse patient subgroups for differential impact of the pandemic.

## 2. Materials and Methods

### 2.1. Data Source and Study Design

The study received approval from the institutional scientific review committee and was exempted from the institutional review board on the basis of the database being deidentified. This study utilized the nationwide, deidentified electronic health record-derived, Flatiron Health database, originating from approximately 280 oncology clinics representing approximately 800 sites of care across the United States. The Flatiron Health database is a longitudinal database, comprising patient-level structured and unstructured data, created via technology-enabled abstraction. This study sought to compare the cohorts of advanced NSCLC patients treated in 2019 (pre-pandemic cohort) and in 2020 (pandemic cohort). Patients in the pandemic cohort initiated treatment between 1 March and 31 July 2020 and was followed through 1 March 2021. To minimize any seasonal trend, the comparator cohort was selected from the exact period in the year immediately before the pandemic. The pre-pandemic cohort consisted of those who initiated treatment between 1 March and 31 July 2019 and was followed through 1 March 2020. To minimize potential effects of the pandemic on the outcome of the pre-pandemic cohort, follow-up was censored on 1 March 2020, which marked the beginning of the community COVID-19 outbreak in the United States.

### 2.2. Cohort Selection and Outcome Measurement

Cohorts were selected from advanced NSCLC patients who started pembrolizumab, with or without chemotherapy, during the specified periods. Patients with alterations in targetable *ALK* or *EGFR* were excluded from analyses since frontline pembrolizumab was not indicated for this population. Furthermore, those with delayed initiation of treatment beyond 120 days from the date of advanced NSCLC diagnosis were excluded according to the recommended database rule. Real-world overall survival is defined as an interval between frontline treatment initiation and last follow-up or death. Since the date of death in this database was available only in a month-year format to protect patient confidentiality, the first date of the month after the death month was utilized as the date of death for calculation purposes. Frontline therapy was defined as the first systemic anticancer treatment received since the diagnosis of advanced NSCLC. The performance status was from the Eastern Cooperative Oncology Group (ECOG). Biomarker information was abstracted from unstructured reports scanned into electronic medical records at each medical practice. Geographic location was classified by the state of residence according to the United States Census Bureau.

### 2.3. Statistical Analysis

Descriptive statistics including median and range were used for continuous variables, and count and proportion were used for categorical variables. The nonparametric Mann Whitney test was utilized to compare continuous variables, while the Pearson chi-square test was used to compare categorical variables. Survival was estimated using the method of Kaplan–Meier. The log-rank test was used to compare the survival between pre-pandemic and pandemic cohorts. Cox proportional-hazards regression models were fitted to derive hazard ratios (HR) as well as the corresponding 95% confidence interval (CI) and to investigate the interaction between variables. The significance level was set at *p* < 0.05, and all *p*-values were two-tailed. Statistical analyses and graphics were carried out using SPSS version 26 (IBM, Armonk, NY, USA) and SAS version 9.4 (SAS Institute, Cary, NC, USA).

## 3. Results

### 3.1. Patient Characteristics and Outcomes

Based on study criteria (Figure 1), data from 2090 patients were available for analyses: 1092 patients in the pre-pandemic cohort and 998 in the pandemic cohort. Both cohorts were comparable regarding patient-, cancer-, and treatment-related characteristics (Table 1). Including both cohorts, the median age of patients was 71 years, with 78% of them having ECOG performance status of 0 to 1. Adenocarcinoma constituted 74% in both groups, and among those with available data on PD-L1 expression, 33% had an expression level ≥50%. Overall, 29% of patients received pembrolizumab monotherapy while 71% of patients received pembrolizumab plus chemotherapy.

At the time of analysis, 935 deaths had occurred. Including both cohorts, the median real-world overall survival was 10.7 months (95% CI: 10.2–11.2 months). There was no statistically significant difference in the survival between the pre-pandemic and pandemic cohorts. In the pre-pandemic cohort, the median survival was 10.4 months (95% CI: 9.8–11.1), compared with 11.1 months (95% CI: 10.4–11.7) in the pandemic cohort, *p* = 0.31.

### 3.2. Pembrolizumab Monotherapy

There were 613 patients treated with pembrolizumab monotherapy: 307 patients in the pre-pandemic cohort and 306 patients in the pandemic cohort. Among them, PD-L1 expression level was available from 542 patients. Most patients who received pembrolizumab monotherapy had a high PD-L1 expression level. The proportion of patients with PD-L1 ≥ 50% was similar across cohorts: 168 out of 267 patients (62.9%) in the pre-pandemic cohort and 173 out of 275 patients (62.9%) in the pandemic cohort.

When including all patients who received pembrolizumab monotherapy in the analysis regardless of their PD-L1 expression level, there was no statistically significant difference in the real-world overall survival between the pre-pandemic and the pandemic cohorts. The median real-world survival was 10.4 months (95% CI: 9.8–11.1) in the pre-pandemic cohort, compared with 11.1 months (95% CI: 10.5–11.7) in the pandemic cohorts, *p* = 0.31. However, when considering subgroups of patients based on their PD-L1 expression level, a significant interaction between the cohort and PD-L1 expression level was found (*p*-interaction = 0.02). Among those with PD-L1 < 50%, the median survival was 10.1 months (95% CI: 8.5–11.7). The survival was significantly better in the pandemic cohort than the pre-pandemic cohort: hazard ratio (HR) 0.64 (95% CI: 0.42–0.96, *p* = 0.0283) (Figure 2A). On the contrary, among patients with PD-L1 ≥ 50%, the median survival was 11.6 months (95% CI: 10.0–13.2). However, the survival was not better in the pandemic cohort than the pre-pandemic cohort (Figure 2B). Additional exploratory analyses on patient subgroups according to histology, performance status, or demographical characteristics did not reveal any significant interactions between the cohort and the explored subgroups (Table 2).

Among the group of patients treated with pembrolizumab monotherapy, a multivariable analysis was performed to identify independent predictors of survival by integrating the variables: PD-L1 expression level, age, diagnosis to treatment time, sex, body mass index, race, ECOG, state location, smoking history, payers, time period, interaction between time period, and PD-L1 expression level, as well as histology. There were 474 patients with a complete dataset remaining in this subgroup analysis. Using a backward stepwise variable selection process, the variables ECOG and body mass index were identified as independent predictors of survival. Patients with ECOG zero or those with body mass index ≥ 30 had a better survival rate when compared with others: HR 0.56 (95% CI: 0.37–0.90) and HR 0.65 (95% CI: 0.46–0.92), respectively. We also investigated the proportion of patients with ECOG zero and body mass index ≥ 30 across the pre-pandemic and pandemic cohorts. There was a higher proportion of patients with ECOG zero in the pre-pandemic than the pandemic cohort: 86.3% vs. 78.8%, *p* = 0.01. However, there was no significant difference in the proportion of those with body mass index ≥ 30: 24.4% vs. 20.9%, *p* = 0.29, respectively.

### 3.3. Pembrolizumab plus Chemotherapy

There were 1477 patients treated with pembrolizumab plus chemotherapy: 785 patients in the pre-pandemic cohort and 692 patients in the pandemic cohort. Of these, PD-L1 expression level was available from 1477 patients. Most of the patients had a low PD-L1 expression level. No significant difference was found in the proportion of patients with PD-L1 ≥ 50% between cohorts: 114 out of 600 patients (19.0%) in the pre-pandemic cohort and 105 out of 529 patients (19.2%) in the pandemic cohort.

Among patients receiving chemotherapy plus pembrolizumab, no significant difference was found in the real-world overall survival between the pre-pandemic and pandemic cohorts. The median survival was 10.7 months (95% CI: 9.9–11.4) and 10.9 months (95% CI: 10.1–11.8) in the pre-pandemic and the pandemic cohorts, respectively, *p* = 0.47. Moreover, we observed no significant interaction between the cohort and PD-L1 expression level (*p*-interaction = 0.70), and the survival in the pandemic cohort was comparable to the pre-pandemic cohort across PD-L1 expression levels (Figure 3A,B).

## 4. Discussion

In this analysis of data derived from diverse oncology practices across the United States, we found that the outcome of patients who underwent frontline pembrolizumab-based treatment was impacted by the pandemic in some subgroup populations. Among those treated with pembrolizumab monotherapy, survival was better in the pandemic cohort than in the pre-pandemic cohort, when the PD-L1 expression level was <50%. However, when the PD-L1 level was ≥50%, the survival was not better in the pandemic cohort. This effect modification by PD-L1 expression level was not observed among those treated with pembrolizumab plus chemotherapy.

In the present study, the absence of survival deterioration in the pandemic cohort as compared with the pre-pandemic cohort seems to be at odds with previous studies that have described an increased mortality among lung cancer patients during the COVID-19 pandemic [15,16]. In other population-based studies of patients with pancreatic, esophageal or colorectal cancers, survival was reportedly worse in the pandemic than the pre-pandemic period [17,18]. Nevertheless, it should be observed that not all patients in the pandemic cohort in our study were infected with SARS-CoV2. In fact, according to historical data from the CDC, perhaps only 15–30% of patients in the pandemic cohort, or approximately 150–300 patients, were infected [4]. Considering a case fatality rate of approximately 22%, it can be estimated that fewer than 70 deaths were directly attributable to the virus [19], constituting a small fraction of the overall 935 deaths in this study. Furthermore, in our study, unlike some previous population-based studies [17,18], we included only the patients who successfully managed to receive their cancer treatment; therefore, we did not consider the excess mortality due to unavailable cancer treatment during the pandemic. While these observations may help explain the absence of survival deterioration among the pandemic cohort in our study, they do not explain the improvement in survival during the pandemic among patients with lower PD-L1 expression treated with pembrolizumab monotherapy.

It is plausible that there is a beneficial effect of SARS-CoV-2 infection in this specific population. Previous literature has suggested a possible synergistic effect of vaccination or viral infection with checkpoint inhibitor immunotherapy. For example, influenza vaccination has been associated with better survival among cancer patients undergoing immunotherapy [20]. It is hypothesized that vaccination can enhance the infiltration of central memory T cells into the tissues, leading to an enhanced anti-cancer immune response [21]. Since the COVID-19 vaccine was not available for most of the pandemic period in this study, it is possible that there was a differential impact of COVID-19 infection on tumors expressing high vs. low PD-L1 during pembrolizumab monotherapy. Moreover, during SARS-CoV-2 infection, investigators have described an upregulation of important immune checkpoint receptors and ligands such as PD-1, CTLA-4, TIM-3, LAG-3, and TIGIT [22]. The immune stimulatory effect due to SARS-CoV-2 infection may have been most beneficial among patients with tumors expressing lower PD-L1 undergoing pembrolizumab monotherapy. On the other hand, the effect may be deleterious among patients with tumors expressing high PD-L1, analogous to the addition of ipilimumab to pembrolizumab monotherapy among this patient population, an experiment that has been found to result in excessive toxicities [23]. Regarding the group of patients treated with pembrolizumab plus chemotherapy in our study, the effect modification by PD-L1 expression was not observed. Since cytotoxic chemotherapy can exert its anti-cancer effect independent of immune response, its use may have obscured the impact of the pandemic.

To our knowledge, this study is the first to provide outcome data specific to pembrolizumab-based treatment for lung cancer before and during the COVID-19 pandemic. The advantage of this population-based study includes its relatively large sample size, therefore allowing adequate statistical power to detect the differential effect of the pandemic among population subgroups. However, the study is limited by its non-randomized design, and the data were obtained during routine clinical care, not a controlled research environment, and thus subjected to delayed or missing data. There was a slightly smaller number of patients in the pandemic cohort, suggesting a delay or difficulty in accessing medical care during the pandemic. The dataset is also limited by missing values, and in the subgroup analysis of patients with a complete dataset, the effect modification of PD-L1 expression level by time period was no longer significant Furthermore, important covariates such as confirmation of COVID-19 infection, comorbidity, or socioeconomic status were unavailable. Nonetheless, these confounders should not result in a differential impact of treatment based on PD-L1 expression status. Finally, we do not have data on treatment-related toxicity or cause of death, which could help explain the findings. Nonetheless, the large number of patients helps mitigate statistical power loss from missing data, and the baseline observable characteristics between the pandemic and pre-pandemic cohorts are well balanced, thus reducing the threat of confounders.

In summary, this population-based study demonstrated an improvement in real-world overall survival during the COVID-19 pandemic among a specific group of lung cancer patients undergoing pembrolizumab-based treatment: those with PD-L1 expression level < 50% treated with pembrolizumab monotherapy. Future studies will be needed to understand the underlying mechanisms contributing to the survival improvement.

## Figures and Tables

**Figure 1 jcm-12-01611-f001:**
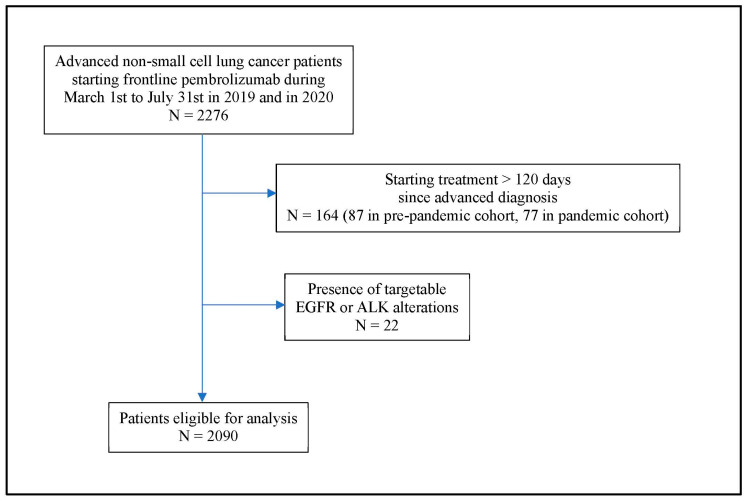
Flow diagram of patient cohort.

**Figure 2 jcm-12-01611-f002:**
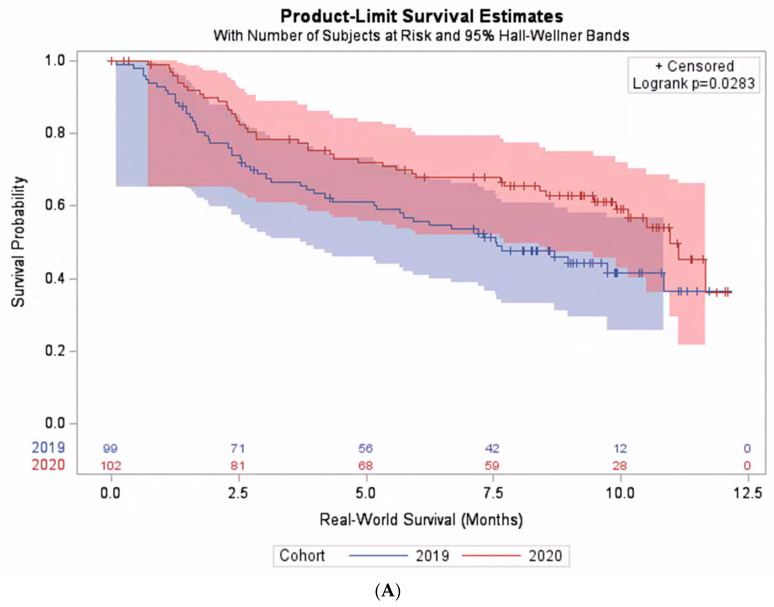
Real-world survival among patients treated with pembrolizumab monotherapy. (**A**) Subpopulation with PD-L1 expression level < 50%. (**B**) Subpopulation with PD-L1 expression level ≥ 50%.

**Figure 3 jcm-12-01611-f003:**
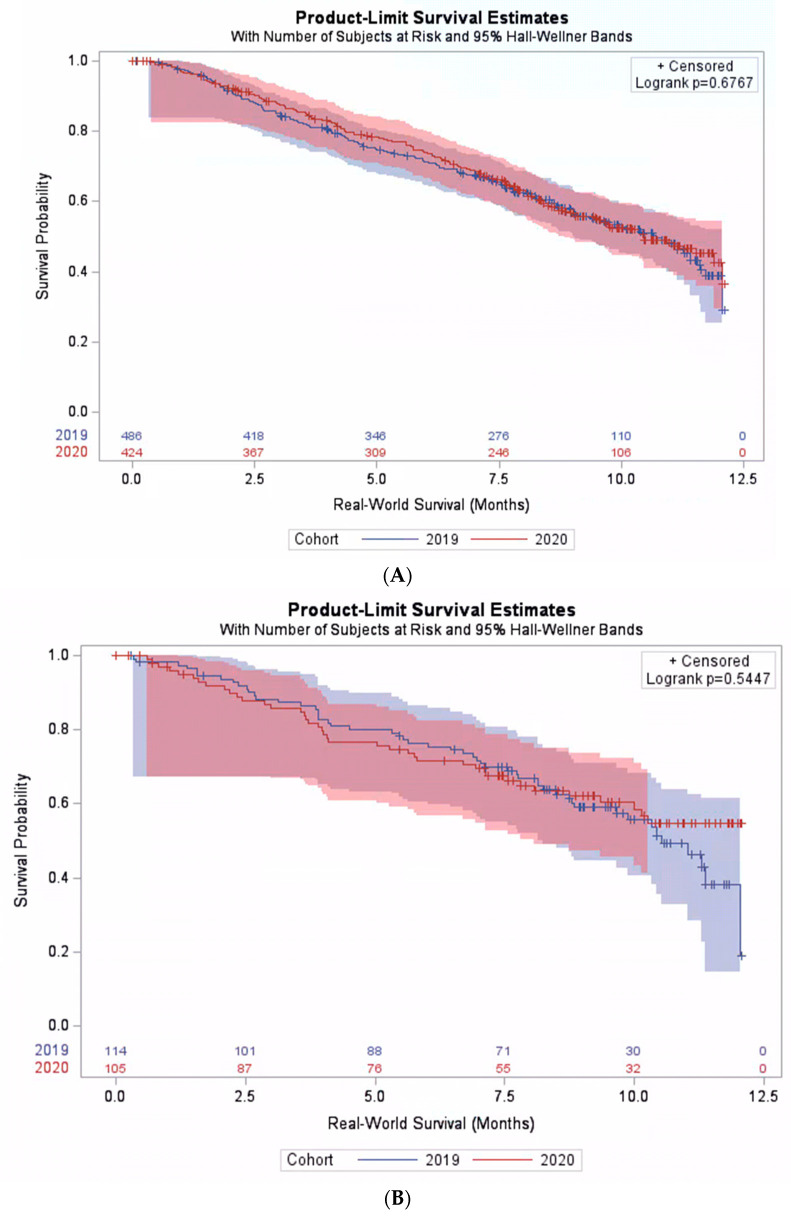
Real-world survival among patients treated with pembrolizumab plus chemotherapy. (**A**) Subpopulation with PD-L1 expression level < 50%. (**B**) Subpopulation with PD-L1 expression level ≥ 50%.

**Table 1 jcm-12-01611-t001:** Patient characteristics.

Characteristics	Pre-Pandemic 2019 Cohort N = 1092 (%)	Pandemic 2020 Cohort N = 998 (%)	All Patients N = 2090 (%)	*p*-Value
Median age (range)	71 (36–83)	71 (30–84)	71 (30–84)	0.17
Median days from diagnosis to treatment (range)	33 (1–120)	32 (0–120)	32 (0–120)	0.32
Sex:				
- Female - Male	497 (45) 595 (55)	452 (45) 546 (55)	949 (45) 1141 (55)	0.92
Median body mass index (range)	25.7 (14.2–71.6)	25.8 (13.7–59.4)	25.7 (13.7–71.6)	0.95
Race:				
- White - Others	740 (77) 224 (23)	654 (74) 232 (26)	1394 (75) 456 (25)	0.14
ECOG *:				
- 0 - 1 - ≥2	207 (28) 370 (50) 163 (22)	233 (32) 348 (47) 157 (21)	440 (30) 718 (48) 320 (22)	0.31
Treatment location ^†^:				
- Northeast states - South states - Midwest states - Western states	206 (21) 167 (17) 472 (49) 120 (12)	164 (18) 145 (16) 483 (53) 125 (14)	370 (20) 312 (17) 955 (51) 245 (13)	0.15
Smoking status:				
- History of smoking - Non-smoker	993 (91) 97 (9)	915 (92) 82 (8)	1908 (91) 179 (9)	0.76
Insurance:				
- Medicaid, charity, unknown - Others	167 (15) 925 (85)	146 (15) 852 (85)	313 (15) 1777 (85)	0.67
Histology:				
- Non-squamous cell - Squamous cell	292 (27) 800 (73)	255 (26) 743 (74)	547 (26) 1543 (74)	0.54
PD-L1 level ^‡^:				
- <1% - 1–49% - ≥50%	304 (35) 281 (32) 282 (33)	279 (35) 247 (31) 278 (34)	583 (35) 528 (32) 560 (33)	0.63
Treatment regimen:				
- Pembrolizumab - Pembrolizumab/chemotherapy	307 (28) 785 (72)	306 (31) 692 (69)	613 (29) 1477 (71)	0.20

* Data available from 1478 patients; ^†^ data available from 1882 patients; ^‡^ data available from 1671 patients.

**Table 2 jcm-12-01611-t002:** Mortality risk among patients in the pandemic cohort as compared with the pre-pandemic cohort in population subgroups.

Subgroup	Pembrolizumab Monotherapy: HR * (95% CI)	*p*-Value ^†^	Pembrolizumab Plus Chemotherapy: HR * (95% CI)	*p*-Value ^†^
PD-L1 expression level:				
- ≥50% - <50%	1.17 (0.85–1.61) 0.64 (0.42–0.96)	0.02	0.88 (0.58–1.33) 0.96 (0.79–1.17)	0.70
Age				
- ≥70 years - <70 years	0.91 (0.68–1.22) 0.88 (0.58–1.34)	0.83	0.91 (0.74–1.13) 0.99 (0.79–1.24)	0.61
Diagnosis to treatment time:				
- ≥30 days - <30 days	1.01 (0.75–1.36) 0.77 (0.52–1.14)	0.29	0.91 (0.73–1.14) 0.99 (0.81–1.24)	0.52
Sex:				
- Female - Male	0.97 (0.69–1.35) 0.88 (0.63–1.24)	0.65	0.88 (0.69–1.12) 0.99 (0.82–1.21)	0.47
Body mass index:				
- ≥30 - <30	0.82 (0.47–1.41) 0.93 (0.72–1.22)	0.73	0.91 (0.66–1.25) 0.96 (0.81–1.14)	0.73
Race:				
- White - Non-white	0.91 (0.68–1.22) 1.01 (0.59–1.70)	0.73	0.93 (0.77–1.12) 0.99 (0.72–1.37)	0.75
ECOG performance status:				
- 0 - ≥1	0.92 (0.71–1.18) 1.26 (0.59–2.66)	0.31	0.99 (0.83–1.17) 0.85 (0.60–1.22)	0.46
State location:				
- Midwest states - Non-Midwest states	1.37 (0.72–2.61) 0.84 (0.65–1.09)	0.17	1.14 (0.77–1.69) 0.92 (0.78–1.08)	0.33
Smoking history:				
- Ever smoked - Never smoked	0.88 (0.68–1.13) 1.44 (0.67–3.08)	0.22	0.97 (0.83–1.14) 0.68 (0.38–1.23)	0.25
Payers:				
- Medicaid, charity, unknown - Other insurances	0.59 (0.32–1.08) 0.99 (0.77–1.29)	0.09	0.82 (0.55–1.23) 0.97 (0.82–1.14)	0.49
Histology:				
- Squamous cell - Other histology	0.79 (0.51–1.24) 0.96 (0.72–1.27)	0.55	1.03 (0.77–1.39) 0.92 (0.77–1.10)	0.51

* HR, hazard ratio of the pandemic cohort with the pre-pandemic cohort serving as a reference group; ^†^ *p*-value for an interaction between the cohort and the variable. Each interaction term was assessed separately, adjusting for other main effects of the covariates.

## Data Availability

Data not available due to restriction by data source.

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
