# Peer review of "Impact of COVID-19 Pandemic on Frontline Pembrolizumab-Based Treatment for Advanced Lung Cancer"

_jcm, 2023, doi:10.3390/jcm12041611_

Round 1

Reviewer 1 Report

Abstract & conclusions:

The results are interesting, but the conclusions reach too far.

For example, you can't say the survival was "numerically worse" within the PDL1 >50% group - the CI straddles 1 with high p-value. The effect just is not there. 

Further - and importantly, you haven't shown that this differential outcome relates to efficacy of immunotherapy. The effect, if real, may relate to differential impact of covid infection, or vaccination, on patients who are high vs low PDL1 expressors. You've said that 15-30% of contemporaneous patients were infected with covid - enough to impact that outcome. Clearly a covariate of infection with covid +/- would be highly valuable - is there anyway to obtain this? If not - this is a weakness of the paper that needs to be addressed. 

Additionally, there are other potential confounders between pre-pandemic / pandemic you have not considered. Behaviour of patients pre/post, treatment cycles received, vaccination status, stage migration (later diagnoses / greater volume of disease at dx during pandemic)

Within introduction:

To date, however, no studies have established a significant change in toxicities of checkpoint inhibitors during SARS-CoV-2 infection or vaccination (13, 14).

--> this doesn't make sense to me.  This study also has not looked at toxicity?

There appear to be a lot more squamous patients in the pandemic and combined cohort (vs non-squam within the table) - have you flipped the numbers?

Author Response

Abstract & conclusions:

The results are interesting, but the conclusions reach too far. For example, you can't say the survival was "numerically worse" within the PDL1 >50% group - the CI straddles 1 with high p-value. The effect just is not there. 

>>Thank you very much. We have revised this statement in section 3.2 and throughout the paper as suggested.

Further - and importantly, you haven't shown that this differential outcome relates to efficacy of immunotherapy. The effect, if real, may relate to differential impact of covid infection, or vaccination, on patients who are high vs low PDL1 expressors.

You've said that 15-30% of contemporaneous patients were infected with covid - enough to impact that outcome. Clearly a covariate of infection with covid +/- would be highly valuable - is there anyway to obtain this? If not - this is a weakness of the paper that needs to be addressed.

>>Thank you very much. We have added the reviewer’s point in the discussion. We have also added that for the most part of the pandemic period in this study, there was no approved vaccine available.

The Flatiron database does not have information on confirmed COVID cases. Although from an ecological standpoint, it is clear that COVID19 infection during the pandemic occurred, we cannot narrow down to the individual-patient level to specify which patient got infected. This weakness has now been acknowledged in the discussion.

Additionally, there are other potential confounders between pre-pandemic / pandemic you have not considered. Behaviour of patients pre/post, treatment cycles received, vaccination status, stage migration (later diagnoses / greater volume of disease at dx during pandemic)

>>Thank you very much. These confounders should not have resulted in a differential impact of treatment based on PD-L1 expression status. We have added this in the discussion.  

Within introduction:

To date, however, no studies have established a significant change in toxicities of checkpoint inhibitors during SARS-CoV-2 infection or vaccination (13, 14).

--> this doesn't make sense to me.  This study also has not looked at toxicity?

>>Thank you very much. We have revised the statement as suggested.

There appear to be a lot more squamous patients in the pandemic and combined cohort (vs non-squam within the table) - have you flipped the numbers?

>>Thank you very much. We have revised the table as suggested.

Reviewer 2 Report

Major Comments:

Pg. 3 – Results Section 3.1 – There were nearly 100 fewer patients within the pandemic cohort as compared to the pre-pandemic cohort even though both cohorts are derived over a similar duration of follow-up.  Please comment on whether this is likely to suggest that certain patients delayed receiving diagnostic care for lung cancer, or if there are other known explanations such as a change in the underlying Flatiron population size.  Related to the latter, it would be helpful to report if the overall size of the Flatiron population differed between the pre-pandemic and pandemic periods to further understand these numbers in context.

Pg. 5 – Results Section 3.2 – The p-values and confidence intervals for the median survival regardless of PD-L1 expression do not seem to jibe.  Normally if each of the point estimates falls outside of the confidence interval for the other value to which it is being compared (as is the case here), the p-value will be < 0.05 and statistically significant.  However, a p-value of 0.46 is reported here and I would suggest re-reviewing the statistical analysis performed.

Pg. 6 – Figure 2 – The figures show survival over time for patients with PD-L1 < 50% that is no worse, and possibly better at some time points, compared to for patients with PD-L1 50%.  This is a surprising finding because the KEYNOTE-042 trial reported relatively lower efficacy and lower absolute survival for patients with PD-L1 <50% as compared to PD-L1 50%.  It is suggested to further evaluate whether the characteristics of the patients in these two groups differs in some way that could explain this finding and possibly relate to the differences in survival observed between time periods for the PD-L1 <50% patients.

Pg. 6 – Figure 2 – It is recommended to also conduct a multivariate analysis to see if the relationship between PD-L1 expression and survival for Pembrolizumab monotherapy persists after controlling for all other confounders.  Although strong relationships are not seen for other confounders, it is possible that collectively there may be some influence exerted on the findings.

Minor Comments:

Pg. 1 – Abstract Methods – Suggest to modify sentence to “The pre-pandemic cohort consisted of those initiating treatment between March and July of 2019.”

Pg. 1 – Background – 1st Paragraph - In the United States, Pembrolizumab monotherapy is approved for patients with PD-L1 1% and not only 50%.

Pg. 2 – Section 2.2 – In discussing the exclusion of patients who initiated treatment > 120 days from diagnosis, I think it would be useful to report in the text the % of patients in each cohort with this type of treatment delay, as a delay in receiving care was a concern related to the pandemic.

Author Response

REVIEWER 2:

Major Comments:

Pg. 3 – Results Section 3.1 – There were nearly 100 fewer patients within the pandemic cohort as compared to the pre-pandemic cohort even though both cohorts are derived over a similar duration of follow-up.  Please comment on whether this is likely to suggest that certain patients delayed receiving diagnostic care for lung cancer, or if there are other known explanations such as a change in the underlying Flatiron population size.  Related to the latter, it would be helpful to report if the overall size of the Flatiron population differed between the pre-pandemic and pandemic periods to further understand these numbers in context.

>>Thank you very much. We have added in the discussion that There was a smaller number of patients in the pandemic cohort, suggesting delay or difficulty in accessing the care during the pandemic. We have confirmed that there was no administrative change in the data structure.

Pg. 5 – Results Section 3.2 – The p-values and confidence intervals for the median survival regardless of PD-L1 expression do not seem to jibe.  Normally if each of the point estimates falls outside of the confidence interval for the other value to which it is being compared (as is the case here), the p-value will be < 0.05 and statistically significant.  However, a p-value of 0.46 is reported here and I would suggest re-reviewing the statistical analysis performed.

>>Thank you very much. We have revised this in the section 3.2 as pointed out.

Pg. 6 – Figure 2 – The figures show survival over time for patients with PD-L1 < 50% that is no worse, and possibly better at some time points, compared to for patients with PD-L1 ≥ 50%.  This is a surprising finding because the KEYNOTE-042 trial reported relatively lower efficacy and lower absolute survival for patients with PD-L1 <50% as compared to PD-L1 ≥ 50%.  It is suggested to further evaluate whether the characteristics of the patients in these two groups differs in some way that could explain this finding and possibly relate to the differences in survival observed between time periods for the PD-L1 <50% patients.

>>Thank you very much. The survival was actually better among patients with PD-L1 ≥ 50%.  We have specified the median survival in the section 3.2. We have also explored the characteristics between groups as suggested (see below).

Pg. 6 – Figure 2 – It is recommended to also conduct a multivariate analysis to see if the relationship between PD-L1 expression and survival for Pembrolizumab monotherapy persists after controlling for all other confounders.  Although strong relationships are not seen for other confounders, it is possible that collectively there may be some influence exerted on the findings.

>>Thank you very much. We have performed a multivariable analysis among patients receiving monotherapy as suggested. We have included this in the section 3.2. In the final model, there were 2 independent variables predicting survival. We have added this results in the text in the result section. We have also compared the variable distribution by cohorts. There was actually a higher proportion of patients with ECOG zero in the pre-pandemic cohort. As such, this could have favored a better survival among the pre-pandemic cohort. However, this is only a subgroup analysis and has to be interpreted with caution.

Minor Comments:

Pg. 1 – Abstract Methods – Suggest to modify sentence to “The pre-pandemic cohort consisted of those initiating treatment between March and July of 2019.”

>>Thank you very much. We have revised this as suggested

Pg. 1 – Background – 1st Paragraph - In the United States, Pembrolizumab monotherapy is approved for patients with PD-L1 ≥ 1% and not only ≥ 50%.

>>Thank you very much. We have made a revision as suggested.

Pg. 2 – Section 2.2 – In discussing the exclusion of patients who initiated treatment > 120 days from diagnosis, I think it would be useful to report in the text the % of patients in each cohort with this type of treatment delay, as a delay in receiving care was a concern related to the pandemic.

>>Thank you very much. We have added in the figure 2 that there were 87 such patients excluded in the pre-pandemic and 77 in the pandemic cohorts.

Round 2

Reviewer 2 Report

Pg. 6 – Figure 2 – It is recommended to also conduct a multivariate analysis to see if the relationship between PD-L1 expression and survival for Pembrolizumab monotherapy persists after controlling for all other confounders.  Although strong relationships are not seen for other confounders, it is possible that collectively there may be some influence exerted on the findings.

Author Response: Thank you very much. We have performed a multivariable analysis among patients receiving monotherapy as suggested. We have included this in the section 3.2. In the final model, there were 2 independent variables predicting survival. We have added this results in the text in the result section. We have also compared the variable distribution by cohorts. There was actually a higher proportion of patients with ECOG zero in the pre-pandemic cohort. As such, this could have favored a better survival among the pre-pandemic cohort. However, this is only a subgroup analysis and has to be interpreted with caution.

Revision Comment: Thank you for performing the multivariate analysis and describing the results in the paper.  It was not clear whether the time period was included in the regression (pre-pandemic vs. post-pandemic) and whether this remained statistically significant after controlling for the covariate differences between the patients in each cohort.  If this was included and is no longer statistically significant, then given that the PD-L1 status was not a significant determinant of survival in the pre- vs. post-pandemic period, the observed survival differences would appear to only be due to differences in other patient covariates like ECOG status and BMI between the cohorts and not due to the patients PD-L1 status.  The discussion should then be revised to reflect, as currently PD-L1 status is described as an effect modifier for pre- vs. post-pandemic survival.

Author Response

Revision Comment: Thank you for performing the multivariate analysis and describing the results in the paper. 

It was not clear whether the time period was included in the regression (pre-pandemic vs. post-pandemic) and whether this remained statistically significant after controlling for the covariate differences between the patients in each cohort. 

>>>Thank you very much. We have now added the time period and the interaction term between PD-L1 level into the model. The analysis still did not retain these variables as significant. We have added this into the results.

If this was included and is no longer statistically significant, then given that the PD-L1 status was not a significant determinant of survival in the pre- vs. post-pandemic period, the observed survival differences would appear to only be due to differences in other patient covariates like ECOG status and BMI between the cohorts and not due to the patients PD-L1 status.  The discussion should then be revised to reflect, as currently PD-L1 status is described as an effect modifier for pre- vs. post-pandemic survival.

>>>Thank you very much. Due to the much smaller number of patients available for this multivariable analysis (many patients have missing ECOG value etc.), the finding from this subgroup analysis can be misleading as important finding can be lost. We have presented the finding objectively in the result section and added in the discussion section for readers to consider the issue with missing values.